# Impact of COVID-19 pandemic related stressors on patients with anxiety disorders: A cross-sectional study

**Till Langhammer** *, **Carlotta Peters, Andrea Ertle, Kevin Hilbert** , **Ulrike Lueken**

Department of Psychology, Humboldt-Universität zu Berlin, Berlin, Germany

* till.Langhammer@hu-berlin.de

**Data Availability Statement:** Data cannot be shared publicly as this is not included in the informed consent by patients. The study was

## Abstract

The COVID-19 pandemic and related containment measures are affecting mental health, especially among patients with pre-existing mental disorders. The aim of this study was to investigate the effect of the first wave and its aftermath of the pandemic in Germany (March–July) on psychopathology of patients diagnosed with panic disorder, social anxiety disorder and specific phobia who were on the waiting list or in current treatment at a German university-based outpatient clinic. From 108 patients contacted, forty-nine patients (45.37%) completed a retrospective survey on COVID-19 related stressors, depression, and changes in anxiety symptoms. Patients in the final sample (n = 47) reported a mild depression and significant increase in unspecific anxiety ($d$ = .41), panic symptoms ($d$ = .85) and specific phobia ($d$ = .38), while social anxiety remained unaltered. Pandemic related stressors like job insecurities, familial stress and working in the health sector were significantly associated with more severe depression and increases in anxiety symptoms. High pre-pandemic symptom severity (anxiety/depression) was a risk factor, whereas meaningful work and being divorced/separated were protective factors (explained variance: 46.5% of changes in anxiety and 75.8% in depressive symptoms). In line with diathesis-stress models, patients show a positive association between stressors and symptom load. Health care systems are requested to address the needs of this vulnerable risk group by implementing timely and low-threshold interventions to prevent patients from further deterioration.

## Introduction

Fear of infection and health concerns, economic and financial uncertainty, social isolation, and familial stress are major challenges in a worldwide pandemic, possibly leading to increased psychological distress and mental health problems [1, 2]. Large parts of societies were affected and many areas of life faced restrictions by the pandemic. Concerns were quickly raised about the "collateral damage" of containment measures on mental health [3]. Only a few weeks after the outbreak of the COVID-19 pandemic, first surveys concerning its immediate psychological impact indicated extensive psychological distress in the general population [4, 5]. The most frequently reported mental health issues during the first wave of the COVID-19 pandemic were symptoms of depression, anxiety, insomnia, and distress [3, 6, 7]. It may be argued that patients with pre-existing mental disorders are particularly at risk for symptom deterioration.

reviewed by the Research Ethics Committee of the Department of Psychology at Humboldt-Universtät zu Berlin, Germany (proposal number: 2020-41) and the Ethics Committee demands those strict data sharing plans in case we include patients from our outpatient clinic. Further, the dataset provides health-related data which is considered particularly sensitive data according to German data protection acts. However, deidentified participant data with annotations will be made available to other researchers upon reasonable request (towards the study administration at coro-na-stressfrei. psychologie@hu-berlin.de).

**Funding:** The article processing charge was funded by the Deutsche Forschungsgemeinschaft (DFG, German Research Foundation) in the form of a grant (#491192747), and the Open Access Publication Fund of Humboldt-Universität zu Berlin in the form of a grant to TL. No additional external funding was received for this study. The funders had no role in study design, data collection and analysis, decision to publish, or preparation of the manuscript.

**Competing interests:** The authors have declared that no competing interests exist.

Regarding risk factors, uncertainties concerning health and economic status increase psychological distress and trigger psychopathological symptoms, anxiety and depression [5, 8, 9–11]. Previous research suggests that social isolation increases vigilance for threat, heightens feelings of vulnerability, decreases emotion recognition ability and causes serious mental health problems [1, 12]. Having children has been associated with poor post-disaster mental health outcomes, probably because of higher stress and responsibility [13, 14]. During the pandemic, families with children responded with greater fear, anxiety and depressive symptoms [15].

The pandemic itself and the restrictions necessary to prevent harm and death are particularly challenging for people already suffering from mental illness [16]. Mental health problems prior to disasters have shown to be a main risk factor for post-disaster psychopathology [13, 17, 18]. In times of the COVID-19 pandemic, individuals with a previous mental illness reported significantly higher depression and anxiety scores, higher distress, and more health anxiety [19–21]. At the same time, access to (mental) health care in Germany was restricted due to the containment measures [22].

Diathesis-stress models assume that the risk for developing symptoms of mental disorders in times of crises heightens due to the interaction between stress and specific vulnerabilities (e.g., anxiety disorders).

Patients with an anxiety disorder are prone to fear of infection and health concerns, economic and financial uncertainty. Anxiety disorders, especially social anxiety disorder and panic disorder, are associated with high intolerance of uncertainty [23–25], whereas health anxiety has been related to patients with panic disorder [26, 27]. Individuals with pre-existing mental illness are especially vulnerable to losing their jobs during economic recessions, especially if they are male and have low levels of education [28]. It seems plausible to assume an association between those factors and an increase in unspecific anxiety and symptoms of social anxiety disorder, panic disorder and specific phobia.

Avoidant and safety behaviours lead to short-term relief, but long-term aggravation: According to the two-factor theory of fear acquisition [29], in the long-term avoidance behaviours maintain anxiety through negative reinforcement [30], thus leading to a vicious circle. Containment measures and the potential risk of infection have "prescribed" avoidance and safety behaviors in the general population and may have reinforced these maladaptive behaviors in patients with anxiety. Especially patients with social anxiety disorder might have been relieved in the short-term in times of a lockdown as most people had only contact to close others. Following the two-factor theory, the short-term relieve would lead to the long-term maintenance of anxiety. In addition to the (the absence of) exposure does not counteract.

Major depressive disorder and dysthymia have shown to co-occur in approximately 20% to 40% of patients with primary anxiety disorders [31]. During the COVID-19 pandemic depressions rank among the most prevalent mental health problems in the general population [32], and pre-disaster mental illness has consistently been found to be a risk factor for post-disaster depression [13]. Brown et al. reported that patients with anxiety disorders have a higher risk of developing a major depression after stressful life events [33]. In a more recent review, the authors concluded that anxiety has a negative effect on depression in terms of symptom worsening, suicidal thoughts, and long-term outcome [34].

Several studies evaluated the accuracy of these assumptions on a level of depressive symptoms and unspecific anxiety [35, 36]. In a study of Pan et al. individuals with a pre-existing mental health condition experienced no change or even a slight decrease in depressive symptoms and unspecific anxiety symptoms during the first lockdown [36]. Another study found a slight decrease in symptoms of depression and no change in anxiety symptoms during the first lockdown for individuals with pre-existing mental health conditions [35]. Several studies

demonstrated higher COVID-19 related stress for individuals with an anxiety disorder and major depression [37, 38]. Bendau et al. could demonstrate the vulnerability of patients with anxiety disorder and depression for increased stress [39]. They investigated changes in COVID-19 related fear, depression, and unspecific anxiety. They used the PHQ-4, which measures anxiety and depression with two items, respectively. They differed between different kinds of anxiety diagnoses as a predictor (measured by asking participants if they had a pre-existing mental disorder). On the one hand, they found higher rates in COVID-19 fear, depression and anxiety individuals claiming to have had an pre-existing anxiety disorder. On the other hand, they observed a greater decrease of symptoms over time in patients versus non-patients.

Their study and others determine pre-existing mental disorders as a specific risk factor for symptom load in times of a lockdown. However, those studies used convenience samples without valid diagnostic procedures. For treatment purposes and prevention programs, findings need confirmation in a well-characterized clinical sample. In addition, a transdiagnostic comparison would be helpful to differentiate between different needs for different kinds of anxiety disorders.

In this study, we focused on patients that have been diagnosed clinically with standardized procedures prior to the pandemic (e.g., panic disorder, social anxiety disorder, and specific phobia). In addition to earlier studies, we evaluated the impact of pandemic related stressors on changes in unspecific anxiety and symptoms underlying panic disorder, social phobia, and specific phobia. First, even though promotion of avoidance might have a relieving effect on all patients, especially on social anxiety, we hypothesized an increase of unspecific anxiety, panic symptoms, social anxiety, and specific phobia. Second, we hypothesized depressive symptoms and the worsening of unspecific anxiety to be associated with health concerns, financial worries and pre-pandemic anxiety symptoms and depression. Amount of treatment was expected to be a negatively associated with less increase of symptoms. We expected pre-pandemic symptom severity to be a vulnerability and therefore a risk factor. Several demographic variables were included in the analyses to test for further protective and risk factors.

## Method

### Participants and procedure

We included patients either waiting for treatment or in treatment for anxiety disorders at the specialized outpatient unit for anxiety disorders at Humboldt-Universität zu Berlin[1] who met the diagnostic criteria for primary diagnosis of panic disorder with or without agoraphobia, social anxiety disorder, or specific phobia. Patients were invited by letter to participate in this study. The clinical diagnoses were based on structured clinical interview for DSM-IV (SKID; [40]) conducted prior to study participation during the initial diagnostic procedure in the outpatient clinic. Individuals who do not fulfil the indication for outpatient treatment (acute suicidal tendency, psychotic disorders, substance dependence) are regularly excluded from admission to the outpatient clinic, and hence from this study.

We contacted all patients that received a primary diagnosis in our specialized unit for anxiety disorders in our outpatient clinic. Nineteen patients from the waiting list and 89 patients in treatment opted in for participating in future studies during the diagnostic process and therefore were contacted via postal mailings in July 2020. Forty-nine patients (response rate = 45.37%) participated. We excluded two patients from the study because they did not have a diagnosed anxiety disorder and were mistakenly contacted during the recruitment (they were in treatment for affective disorders). Participants provided written informed consent to take part in the study. We performed the study in accordance with the Declaration of

Helsinki, received approval by the ethics committee of the Department of Psychology the Humboldt University of Berlin (2020–41) and pre-registered the study protocol online on drks.de (DRKS00022280).

## Measurements

Patients completed a set of self-generated questions also used in a parallel survey from our group conducted in the general population [41] regarding health concerns and uncertainties, social isolation, financial worries, increased parenting, and work demands. In addition, we assessed depressive symptoms and changes in symptomatology of panic disorder, social anxiety disorder, and specific phobia. For the latter, we used well-established psychometric tests for anxiety symptoms, but changed the instructions slightly in a way that changes in symptom intensity were addressed. We also used routine data from the last routine diagnostic prior to the outbreak of the pandemic. For more detailed information and psychometrics of the questionnaires used see Supplement 1 in S1 File.

## Demographic information

Patients had completed a general demographics questionnaire in the initial examination in the outpatient clinic including age, sex, education level, employment status, individual and household income, and Socio-economic status (SES).

## Pre-pandemic symptom severity

Severity of anxiety symptoms before the pandemic was assessed with the Hamilton Anxiety Rating Scale (HAM-A; [42]), a clinician-based questionnaire consisting of 14 items with sufficient internal reliability [43]. A sum score of less than 18 indicates mild, 18 to 24 mild to moderate, 25 to 30 moderate to severe, and more than 30 severe anxiety symptoms. Severity of depressive symptoms before the pandemic was assessed with the revised version of Beck Depressive Inventory (BDI-II; [44]), a well-validated self-assessment with 21 items based on the DSM-IV [45].

**Assessment of pandemic-related stressors.** Patients completed questions on four domains: health concerns (5 items), social isolation (3 items), financial worries (4 items), and family- and work-related stress (11 items) based on a 5-point Likert scale (ranging from *not at all* to *a lot/very much*) or with *yes* or *no*. For details and assignment of items to categories, see Figs 3 and 4.

**Assessment of current depressive symptoms.** Severity of depression was measured with the Patient-Health Questionnaire (PHQ-9; [46, 47]). PHQ-9 sum scores of 5, 10, 15, and 20 are associated with mild, moderate, moderately severe and severe depression. The PHQ-9 was chosen for comparability to other studies in the pandemic.

**Assessment of changes in anxiety symptoms.** We adapted questionnaires used during the routine diagnostics to retrospectively assess changes in symptomatology during the lockdown. For most items, we only changed the dimension of the Likert scale. We had to change some items grammatically for a meaningful adaptation of the dimensions of the Likert scale. Patients were asked to rate changes in anxiety and avoidance symptoms on a 5-point Likert scale from -2 ("a lot less") to +2 ("a lot more") and to refer to the period since mid of March 2020 in their answers. We added an additional answer "not applicable", as we were only interested in changes of symptoms that were present before the pandemic. The 72 items were based on three established and validated questionnaires for anxiety disorders: The Liebowitz Social Anxiety Scale (LSAS; [48, 49]), Panic and Agoraphobia Scale (PAS; [50]), and the DSM-5 Severity Measure for Specific Phobia–Adult [51]. We will refer to those modified

questionnaires as LSAS-m, PAS-m and SMSP-m. We omitted two items from the Panic and Agoraphobic Scale. One item was omitted because it is an additional item that is not included in the total score ("Did most attacks occur unexpected or expected [in dreaded situations]?"), the other because it did not fit to the rating scale and could not capture a change in the severity of a specific phobia symptom ("How important were the situations you avoided?"). Thus, the modified version of the PAS (in the following called PAS-m) consisted of 12 items. All scales were recoded to 1 to 5 for the analyses.

### Data analysis

To test the first hypothesis that patients experienced an increase of symptoms of unspecific anxiety, panic symptoms, social anxiety and specific phobia one sample one-sided $t$-tests ($H_0$: $μ < = 3$) were conducted. We calculated means for the LSAS-m, PAS-m and the SMSP-m, excluding items coded as "not applicable" (e.g. not relevant for the patients' symptom profile). We also calculated an overall score of anxiety, which was the average of the previous mean scores.

For the second hypothesis and testing for further protective and risk factors, we conducted two forward stepwise regression analyses. The first one tested the hypothesis that COVID-19 related stress (23 items), demographic (socioeconomic status, marital status, education, job, sex and age) and clinical characteristics (amount of sessions during the pandemic, pre-pandemic anxiety and depression) affect changes in severity of the anxiety symptoms, using the computed overall mean anxiety score. The second one tested the same hypothesis on depressive symptoms. Analyses were carried out using SPSS 22.0 (IBM SPSS Statistics, New York, NY, United States). A significance level of $α < .05$ was considered to indicate statistical significance.

## Results

### Sample characteristics

Details regarding the demographic and clinical characteristics of the sample are presented in Table 1.

### Changes in anxiety symptoms

Patients reported a significant increase in panic symptoms (PAS-m: $t(44) = 4.70$, $p < .001$, $d = .85$) and specific phobia (SMSP-m: $t(40) = 2.4$, $p = .011$, $d = .38$). Social anxiety symptoms did not increase significantly. Unspecific anxiety also increased significantly ($t(46) = 2.26$, $p = .014$, $d = .41$). See Fig 1 for distribution. The effect sizes for the overall increase in anxiety and phobic symptoms can be considered small [52], large effects were observed for panic symptoms.

### Depressive symptoms

On average, patients reported depressive symptoms indicating a mild depressive syndrome (Fig 2). ($M = 8.30$, $SD = 5.57$).

### Pandemic stressors moderating changes in anxiety symptoms and severity of depression

Figs 3 and 4 depict severity of the 23 individual stressors of the pandemic. Two multiple forward regression analyses were conducted to test if the 23 stressors of the pandemic, demographics and clinical characteristics significantly predicted patients' changes in anxiety

**Table 1. Demographic and clinical characteristics of participants (N = 47).**

| Demographics | n | (%/) |
|---|---|---|
| Female gender | 28 | (60%) |
| Age, mean (sd) | 37.30 | (10.78) |
| Marital status | | |
| • Single | 14 | (30%) |
| • Married/Partnership | 31 | (66%) |
| • Separated/Divorced | 2 | (4%) |
| Employment status | | |
| • Unemployed | 5 | (11%) |
| • Employed | 31 | (66%) |
| • In training | 6 | (13%) |
| • Other | 5 | (11%) |
| • Work in Health Sector | 6 | (14%) |
| Housing condition | | |
| • Alone | 9 | (19%) |
| • With partner | 23 | (50%) |
| • With children | 12 | (26%) |
| Socio-economic status, mean (sd) | 10.51 | (3.39) |
| • Low SES | 11 | (23%) |
| • Medium SES | 26 | (55%) |
| • High SES | 6 | (13%) |
| Clinical characteristics | | |
| Diagnosed anxiety disorders | | |
| • Panic disorder with/without agoraphobia | 29 | (62%) |
| • Social anxiety disorder | 21 | (45%) |
| • Specific phobias | 18 | (38%) |
| • Comorbid depression | 23 | (49%) |
| Pre-pandemic symptom severity (HAM-A), mean (sd) | 22.47 | (8.46) |
| Pre-pandemic symptom severity (BDI-II), mean (sd) | 18.42 | (11.84) |
| Number of comorbid diagnoses | | |
| • None | 13 | (28%) |
| • 1 | 13 | (28%) |
| • 2 | 16 | (34%) |
| • ≥3 | 5 | (11%) |
| In treatment | | |
| • Sessions since 15th of march, mean sd | 9.65 | 8.83 |
| • Yes | 40 | (85%) |
| • No (on a waiting list for treatment) | 7 | (15%) |

Clinical characteristics were assessed before the pandemic. HAM-A = Hamilton Anxiety Scale, BDI = Beck Depression Inventory II.

symptoms (average across all changes of anxiety symptoms) and severity of depression (PHQ-9 sum). The regression statistics are given in Table 2.

We exploratory repeated the analysis after excluding all dichotomous items with less than 10% of the participants answering the less likely characteristic (see S1 File).

The results of the first regression indicated four predictors explaining 46.5% of the variance ($R^2 = .47$, $F(4,42) = 9.12$, $p < .001$). Two pandemic related stressors affected changes in

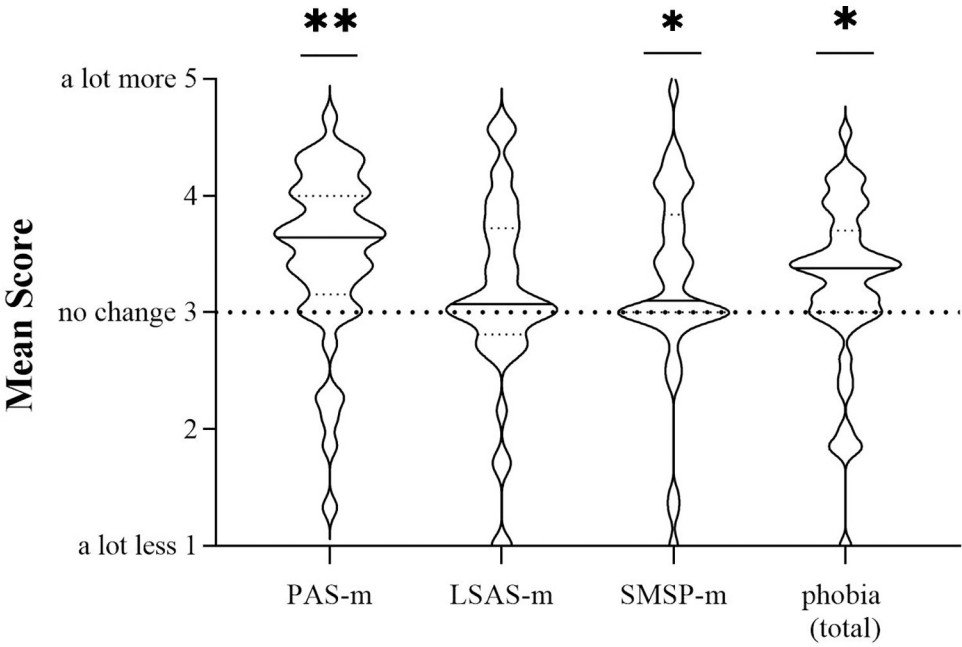

**Fig 1. Changes in anxiety symptoms during and after the first wave of the pandemic.** Mean scores of changes in symptoms of panic disorder, social anxiety disorder and specific phobia. Significance levels indicate difference from a mean score of three, which would be no change. The solid line within the violin plots illustrates the median, and the dashed lines show the interquartile range. ***p < .001. **p < .01. *p < .05.

anxiety: worries about somebody's own work situation significantly predicted changes in anxiety symptoms (β = .35, p = .005), as did overextension with caring/schooling of children (β = .25, p = .046).

The results of the second regression indicated four predictors explaining 75.8% of the variance ($R^2$ = .76, F(4,42) = 32.88, p < .001). While pre-pandemic depression was a significant predictor of symptom change (β = .62, p < .001), three additional pandemic related stressors were included: worries about long-term damage on social relationships significantly predicted depressive symptoms (β = .40, p < .001), as did working in the health sector (β = .26, p = .007), and the subjective meaning of someone's own work (β = -.20, p = .014).

Demographic and clinical characteristics prior to the pandemic were also included in the multiple regression analyses. The higher pre-pandemic anxiety symptom severity, the more pronounced was the reported increase in symptoms of anxiety (baseline HAM-A–Change Score; β = .38, p = .004). In addition, being divorced or separated turned out to be a protective factor (β = -.20, p = .035). In the exploratory analysis, we excluded the items divorced/separated and as an effect of this overextension with caring/schooling of children was not significant after the first two steps of the regression anymore.

For the severity of depressive symptoms, only one risk factor showed incremental validity. Pre-pandemic symptom severity (baseline BDI-II–pandemic PHQ-9; β = .62, p < .001) turned out to predict symptom severity of depression during and after the first wave of the pandemic.

## Discussion

The aim of this study was to explore the impact of the first wave of the pandemic on symptom aggravation in a clinically well-characterized sample of patients diagnosed with a primary

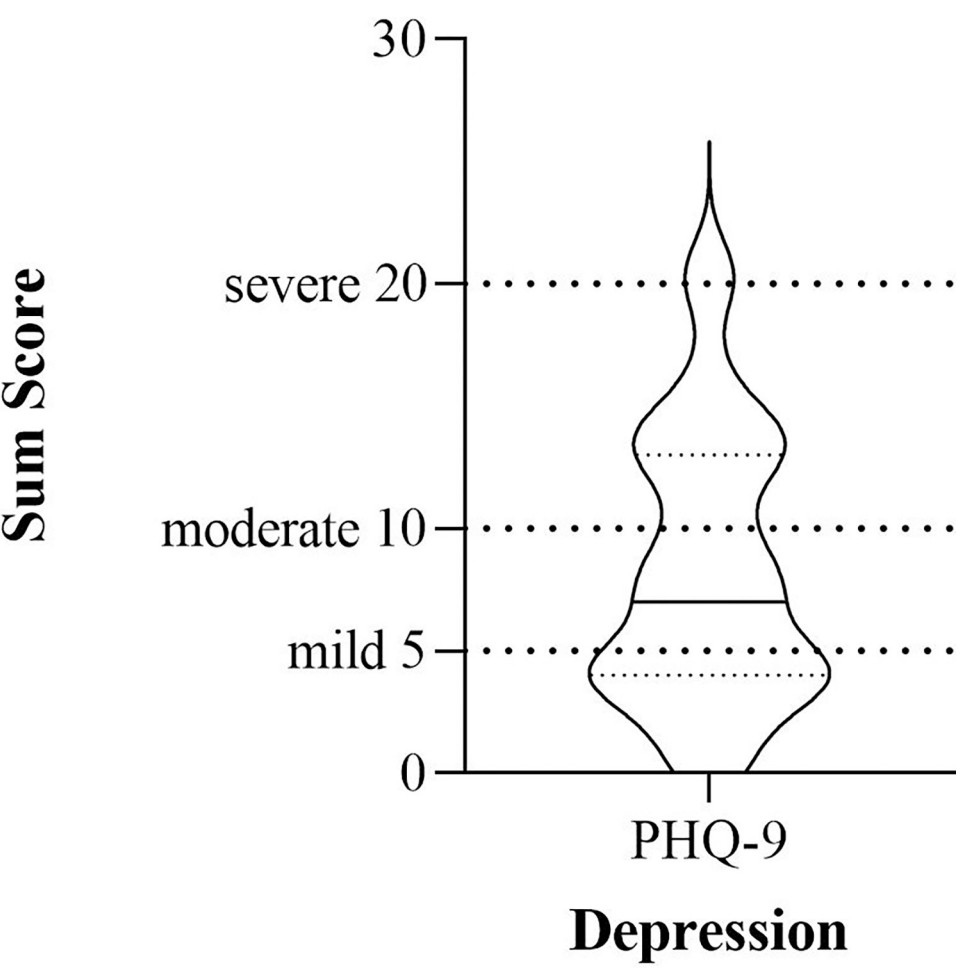

**Fig 2. Severity of depression symptoms during and after the first wave of the pandemic.** The solid line within the violin plots illustrates the median, and the dashed lines show the interquartile range.

anxiety disorder who were in treatment or on a waiting list in a German academic outpatient clinic. Findings indicate a specific increase of unspecific anxiety, panic symptoms and specific phobia symptoms, but not symptoms of social anxiety. In line with general diathesis-stress models, pandemic related stressors like worrying about one's own job and stress through caring/schooling children led to a worsening of anxiety symptoms. Depressive symptoms (a mild depression on average) were associated with worrying about consequences for relationships, working in the health sector and meaningful work during the pandemic. In addition, demographic and clinical characteristics prior to the pandemic turned out to be important. High pre-pandemic symptom severity was identified as a risk factor for both more severe anxiety and depressive symptoms during the pandemic. Surprisingly, being divorced/separated served as a protective factor.

Previous studies [36, 37] indicated a pronounced stress reactivity in patients with a pre-pandemic mental health condition, which then decreased fast. In contrast, our study shows an increase in symptoms, related to stress after a longer period, measured late in summer 2020. However, there are several important differences to our study: we relied on diagnoses assigned by trained clinicians with structured clinical interviews prior to the pandemic (other studies relied on self-reports about pre-existing disorders), we focused on patients with a primary

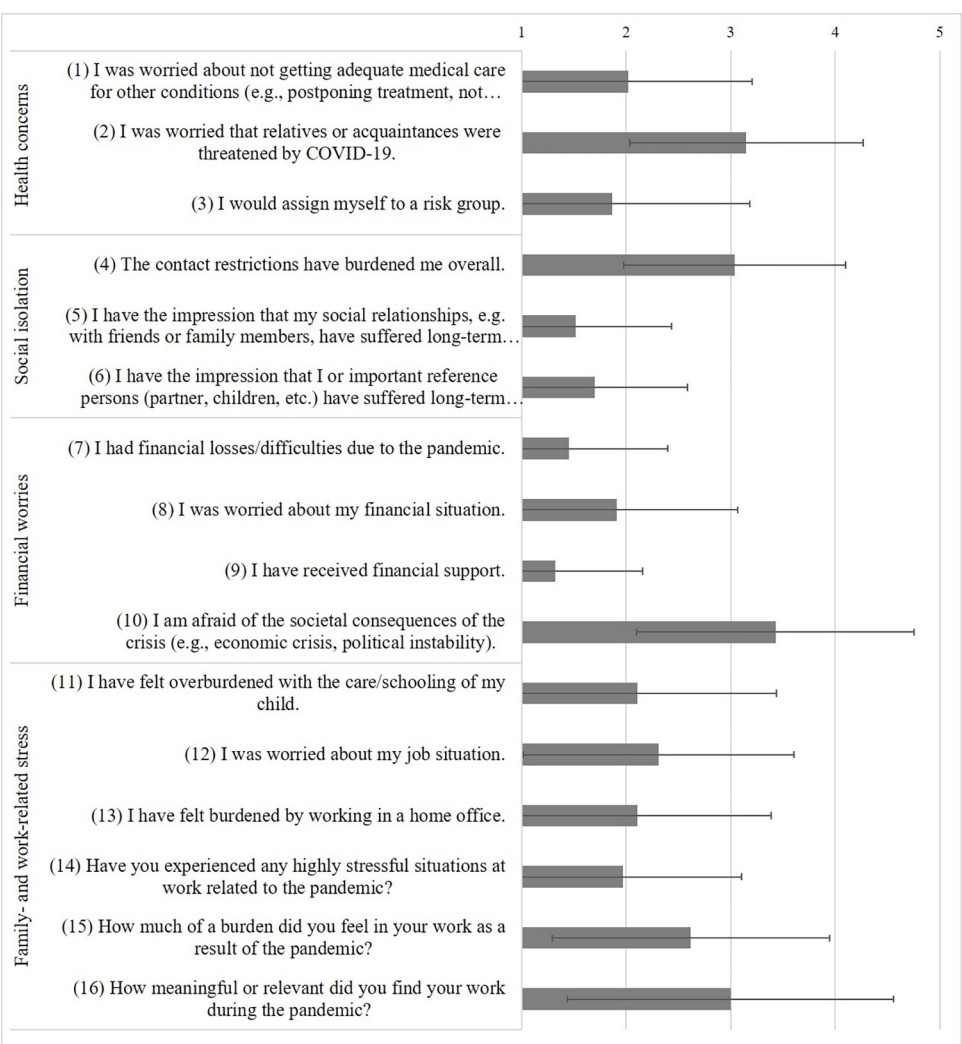

**Fig 3. Questions concerning strength of stressors related to the pandemic.** Mean and standard deviation of all participants.

anxiety disorder, we assessed alterations in symptoms of social anxiety disorder, specific phobia, and panic disorder based on validated questionnaires and we incorporated a longer stress period.

Diathesis-stress models emphasize the interaction between vulnerabilities like mental disorders and stressors like the pandemic and its countermeasures. Therefore, we can assume an increase in symptoms in vulnerable groups. The results of our study confirm this assumption. However, this does not pertain to all diagnostic groups included in the present study.

Anxiety disorders especially differ concerning the stimulus that leads to intensive fear and anxiety and their cognitions that lead to a different trigger of avoidance behaviour and coping strategies, which in turn might be important for the explanation of the differential effect in this study. The decline of social interactions might have led to a temporary reduction of distress and self-deprecating metacognitions particularly in patients with social anxiety disorder, as Morrissette observed in children and adolescents with social phobia during school closures [53]. Social avoidance behaviours might therefore be a reason for the "advantage" of patients with social anxiety disorder compared with those suffering from panic disorder and specific

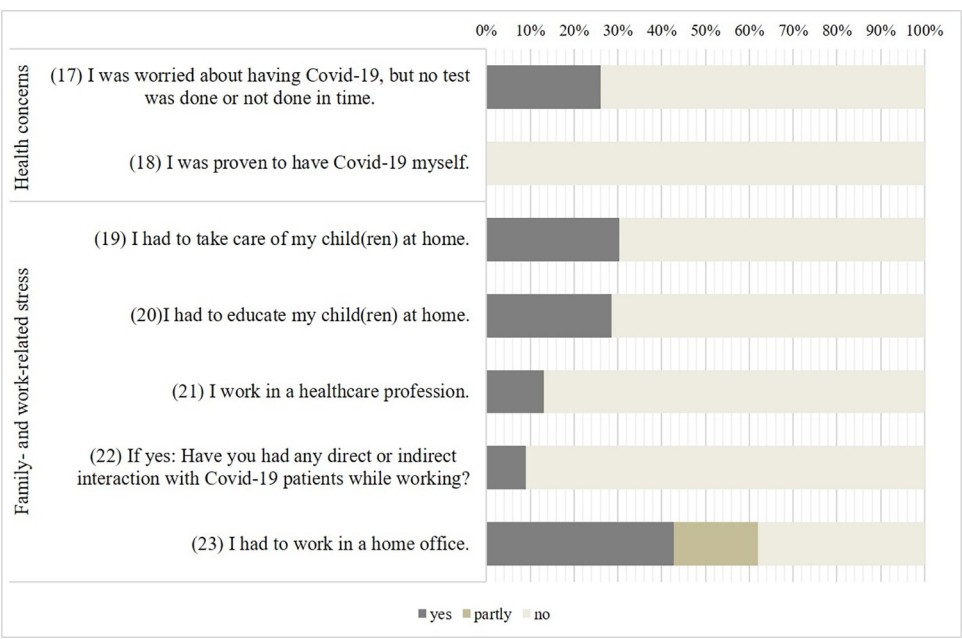

**Fig 4. Questions about the relevance of stressors related to the pandemic.** Percentages of participants answering yes or partly.

phobia. Yet, concerns are raised that reduced exposure to social interactions will lead to increased strain and destabilization in the aftermath of the pandemic when social situations can no longer be avoided [54]. This should be addressed by future studies on the medium- to long-term consequences of the pandemic.

Exogenous stressors–like the challenging life circumstances during a public health emergency–are at least partially responsible for the high prevalence of stress-related mental disorders like anxiety, mood disorders, or post-traumatic stress disorder around the world [55, 56]. The crucial role of stress on mental health in the pandemic is also evident in patients with pre-existing anxiety disorders. Greater COVID-19 related stress was associated with greater increase in symptoms of anxiety and severity of depression. Those stressors, particularly job insecurities and parental stress, combined with pre-pandemic symptom severity could explain approximately half of the variance in changes in anxiety symptoms. In a similar vein, work demands and worries about relationships negatively affected the development of depressive symptoms. Work is an essential resource in everyday life and promotes mental health. In addition, family is supposed to be a resource and social support. Both, work circumstances and familial interactions changed for many people in a stressful manner. Parents had to work from home and at the same time support their children with home schooling. From a clinical perspective, reducing existential stressors by corona aid packages given by governments are not only protecting the economy but may also promote mental health. Health-economic evaluations could help in quantifying the impact of corona aid given by governments on mental health and life quality.

Stressful negative life events, including interpersonal conflicts, financial hardships, or illness, often precede the onset of anxiety disorders [57]. There is less evidence on whether chronic stress contributes to maintaining or increasing anxiety disorders. Wade et al. report that patients with panic disorder with agoraphobia had less of an improvement from treatment when experiencing chronic stress [58]. The results of our study fit this picture, confirming the

**Table 2. Summary of hierarchical regression analysis for variables predicting changes in anxiety symptoms and severity of depression.**

| | B | t | sr² | R | R² | ΔR² |
|---|---|---|---|---|---|---|
| **Changes in anxiety symptoms** | | | | | | |
| **Step 1** | | | | .48 | .23 | .23 |
| Job worries (12) | .48 | 3.65** | .23 | | | |
| **Step2** | | | | .59 | .35 | .13 |
| Job worries | .38 | 3.05** | .14 | | | |
| Pre-pandemic anxiety | .37 | 2.91** | .12 | | | |
| **Step 3** | | | | .64 | .41 | .06 |
| Job worries | .38 | 3.11** | .13 | | | |
| Pre-pandemic anxiety | .45 | 3.50** | .17 | | | |
| Divorced/separated | -.25 | -2.05* | .06 | | | |
| **Step 4** | | | | .68 | .47 | .06 |
| Job worries (12) | .35 | 2.94** | .11 | | | |
| Pre-pandemic anxiety | .38 | 3.02** | .12 | | | |
| Divorced/separated | -.26 | -2.19* | .06 | | | |
| Caring/schooling children (13) | .25 | 2.05* | .05 | | | |
| **Severity of depression** | | | | | | |
| **Step 1** | | | | .75 | .56 | .56 |
| Pre-pandemic depression | .75 | 7.57*** | .56 | | | |
| **Step2** | | | | .83 | .68 | .12 |
| Pre-pandemic depression | .67 | 7.67*** | .36 | | | |
| Relationship worries (5) | .36 | 4.14*** | .12 | | | |
| **Step 3** | | | | .85 | .72 | .04 |
| Pre-pandemic depression | .62 | 7.25*** | .34 | | | |
| Relationship worries (5) | .37 | 4.45*** | .13 | | | |
| Work in health | .20 | 2.38* | .04 | | | |
| **Step 4** | | | | .87 | .76 | .04 |
| Pre-pandemic depression | .62 | 7.69*** | .34 | | | |
| Relationship worries (5) | .40 | 5.09*** | .15 | | | |
| Work in health | .26 | 2.85** | .05 | | | |
| Meaningful work (16) | -.20 | -2.56* | .04 | | | |

$N = 47$

*$p < .05$.

**$p < .01$.

***$p < .001$. The numbers (in parentheses) are corresponding to items in Figs 3 and 4.

relation between stress and increases in anxiety symptoms, and furthermore underline that this patient group may be particularly vulnerable to the psychosocial consequences of a pandemic.

To facilitate coping with stress and to mitigate its effect on mental health, it is important to identify protective factors. Factors potentially buffering the effect of the COVID-19 pandemic on mental health in the general population include availability of up-to-date information, a stable and secure income, self-efficacy, and a positive appraisal style [5, 59–61]. Results of our study indicate that meaningful work can mitigate the effect of the pandemic on depressive symptoms of individuals with anxiety disorders. However, no other assumed protective factors gained significance in mitigating increases in anxiety symptoms. In contrast, being divorced or separated, anticipated as a risk factor, turned out to be protective. As only two participants were divorced or separated, this finding should be regarded with caution.

### Strength and limitations

Although presenting a well-defined clinical sample, results presented in this study are based on a limited number of patients and may not generalize to other anxiety disorders. It may however be assumed that patients with a general anxiety disorder might be particularly vulnerable, as worries about negative consequences are the main symptom of general anxiety disorder and a main reason for increases of symptoms in our study.

Second, although pre-pandemic information about the symptom severity was available, a direct comparison of pre-pandemic and pandemic symptom severity was not meaningful. The period between measurement of pre-pandemic symptom severity and pandemic onset varied substantially. As most of the participants were in treatment this period was confounded with possible symptom changes through treatment. Still, we believe it is a straightforward approach to include both, those in treatment and patients on the waiting list as the pandemic affected the treatment and being on the waiting list. Treatment in some cases switched to online sessions or was disrupted for a longer period because of the restrictions and missing resources for online sessions. Numbers of sessions that varied extensively did not contribute to changes in our statistical model. In some way, the worsening of symptoms seemed to be independent of treatment as usual. Specific interventions in times of crisis seem necessary.

We asked patients to focus on perceived changes in symptomatology due to the pandemic only. Thus, the study has the usual limitation of a cross-sectional study and retrospective assessment (e.g. recall bias): We cannot assume a causal relationship between the variables stress and psychological outcome. Longitudinal studies will have to confirm the findings of this study and replicate the effect of long-term stressors of the pandemic on anxiety and depressive symptoms.

Third, the small sample size in combination with a relatively large amount of predictors in the regression analyses may bear a tendency towards overestimation of explained variance. In our explorative analysis, two predictors, namely being divorced/separated and overextension with caring/schooling of children did not play a role in the best fitting model anymore. These variables should be interpreted with caution.

However, in contrast to other COVID-related studies, we here provide clinical data from actual patients and not a convenience sample from the population. Rendering the data applicable and meaningful to the clinical context. Another strength of this study is that we investigated change in specific symptoms of anxiety disorders rather than examining anxiety in general.

## Conclusion

We found that individuals with anxiety disorders experienced a pronounced increase in anxiety and depressive symptoms, which was largely explained by COVID-19 pandemic related stress. Of note, this was not applicable to social anxiety, possibly indicating a short-term relief due to "prescribed" containment measures.

This study highlights the psychological impact of the pandemic and its psychosocial consequences on the vulnerable group of anxiety disorders that represent the largest group pf mental disorders in the general population and in outpatient treatment. It further emphasized the prominent role of stressors caused by the pandemic for symptom aggravation. Evidencing the validity of the vulnerability-stress-model for the current situation, a pronounced and sustained increase in patients with new incident or relapsing mental disorders should be anticipated, which is also demonstrated by current epidemiological data on a sharp global increase in the prevalence of depressive and anxiety disorders [7]. Health-care systems are required to provide timely and efficient treatment options either for the indicated prevention of mental disorders,

their acute treatment or relapse prevention. Further research is required to track the psychological long-term impact of the pandemic for patients and societies.

## Supporting information

**S1 File.**
(DOCX)

## Author Contributions

**Conceptualization:** Till Langhammer, Carlotta Peters, Andrea Ertle, Kevin Hilbert, Ulrike Lueken.

**Data curation:** Till Langhammer, Carlotta Peters, Andrea Ertle.

**Formal analysis:** Till Langhammer.

**Investigation:** Till Langhammer.

**Methodology:** Till Langhammer, Carlotta Peters, Andrea Ertle.

**Project administration:** Till Langhammer.

**Supervision:** Till Langhammer, Andrea Ertle, Kevin Hilbert, Ulrike Lueken.

**Writing – original draft:** Till Langhammer.

**Writing – review & editing:** Till Langhammer, Kevin Hilbert, Ulrike Lueken.

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
