## [Decision Letter · Decision Letter 0]

28 Mar 2022

PONE-D-21-37020Impact of COVID-19 pandemic related stressors on patients with anxiety disorders: A cross-sectional studyPLOS ONE

Dear Dr. Langhammer,

Thank you for submitting your manuscript to PLOS ONE. After careful consideration, we feel that it has merit but does not fully meet PLOS ONE’s publication criteria as it currently stands. Therefore, we invite you to submit a revised version of the manuscript that addresses the points raised during the review process.

Although your manuscript was found to be of interest, I am unable to accept it in its current form.  The manuscript required major revisions that directly address the concerns of the reviewers. Specifically, the authors should improve the abstract, the introduction and clarify some methodological aspects. 

We look forward to receiving your revised manuscript.

Kind regards,

Serena Scarpelli

Academic Editor

PLOS ONE

Journal Requirements:

Reviewers' comments:

Reviewer's Responses to Questions

**Comments to the Author**

1. Is the manuscript technically sound, and do the data support the conclusions?

Reviewer #1: Partly

Reviewer #2: Yes

Reviewer #3: Yes

2. Has the statistical analysis been performed appropriately and rigorously? 

Reviewer #1: Yes

Reviewer #2: Yes

Reviewer #3: Yes

3. Have the authors made all data underlying the findings in their manuscript fully available?

Reviewer #1: No

Reviewer #2: No

Reviewer #3: No

4. Is the manuscript presented in an intelligible fashion and written in standard English?

Reviewer #1: Yes

Reviewer #2: Yes

Reviewer #3: Yes

5. Review Comments to the Author

Reviewer #1: The authors investigated the impact of COVID-19 pandemic-related stressors on patients with anxiety disorders. Main limitation is a small sample size (49 patients). To find protective or risk factors for symptom severity for anxiety disorder, substantial changes in research methods are needed.

The abstract should include more specific results, including actual values and percentages.

Page 7

We contacted 19 patients from the waiting list and 89 patients in treatment via post in July 2020. Forty-nine patients (response rate = 45.37%) participated. We excluded two patients from the study because they did not have a diagnosed anxiety disorder.

- Patient selection is unclear. It should be clear how many patients were initially selected. Why were patients other than those with anxiety disorders included first?

Reviewer #2: The study looks at the impact of the pandemic on patients with anxiety disorders. The study thus contributes to a better understanding of the impact on this already vulnerable group and to identifying appropriate therapeutic implications.

However, the manuscript has some weaknesses, which I would like to point out below with questions for the authors.

Introduction in general: It would be great for the importance of the manuscript if the authors emphasised again at the end of the introduction why the question is important/novel.

Line 116: why were 19 people from the waiting list contacted. Then 7 people were included in the calculations. Wouldn't it be important to exclude these persons, since they had no therapeutic support during this time and thus differ substantially from the other persons in terms of support during this time? How do the authors justify these 7 persons or why were these 7 persons included?

Line 141: I am still not entirely clear whether the individual domains of the stressors should be considered coherently or whether the items simply stand alone and thus a statement on reliability is not necessary.

If one looks at the individual items, then for some questions it could also lead to an item having to be answered with "does not apply" (e.g. “worried about job situation” cannot be answered if one is already unemployed. “schooling of my child” cannot be answered if one does not have a child). Was there such an answer category? if not, how do the authors deal with the fact that some items, which are also later in the regression, could possibly not be answered correctly?

Line 149 ff: can the authors say something about the validity and reliability of the adapted questionnaires? The answer category was changed for this study and it would be good for the reader if there were more psychometric information.

This is especially important for the PAS-m, where 2 items were also removed.

Line 222 ff: it is not clear to me how the 23 stressors were included in the regression. The figures show only 4 at a time and it is not clear from the text how they were chosen and also why this order was chosen

e.g. in the first regression one would expect “pre-pandemic anxiety” to be included first.

If all 23 predictors were included, this is also not permissible due to the sample size.

It would be imperative for the understanding and interpretation of the results if the authors would present this regression in a little more detail or explain how they proceeded here (which predictors, why this order).

It would also be very welcome if it were clear to the reader exactly which predictors in the regression correspond to which items from Figures 3a and 3b. This is not possible at the moment.

Line 258/259: I am not sure how profound this statement is, as only 2 people were divorced/separated at all. This has already been included in the limitations, but this seems insufficient to me under the fact that it is so small, as it is strongly shown as a protective factor here.

Line 342 ff: The small sample size seems to be a limitation as well as the fact that it is a sample that was already in treatment. It is important to mention this, especially because the authors point out the practical implications that treatment options should be provided. However, this sample was in treatment and the authors should discuss again in detail what effect the treatment had on the increase. Did the therapists already react to this in treatment? How long were the individuals in treatment for, or how often were there therapy sessions during this time, or was therapy interrupted and this could have been an explanation for the increase?

Minor comments:

In line 76: would it be possible to find a more recent source?

Lines 79, 83: According to the journal's guidelines, is the year given in this form of citation? Otherwise, only a number is given here.

Line 90: Here it would be helpful to know in whom higher rates were found.

Line 108/109: it would be great for understanding if the German designation could be translated.

Line 194: it would also be interesting to know the burden of the patients at the time of admission with regard to anxiety symptoms in order to get an idea of how burdened the persons were.

Table 1:

Under the subcategory "Employment status", was there also a question about working in the "health sector" or did this answer come from the stressors?

Under the upper category "Domiciliary status", only the category "with partner" and "with children" was possible, or could one also tick "with partner and children"?

Under this category, 100% does not come up either.

In the upper category SES, the last line says (3.39%). Something seems to have slipped here.

Figure 2:

The first wave was already in March/April and therefore already over in July. It is therefore difficult to say that this was the symptom severity during the first wave (as described in the title of the figure). In July, the incidences had already been very low.

Reviewer #3: The manuscript "Impact of COVID-19 pandemic related stressors on patients with anxiety disorders: A cross sectional study" presents an interesting investigation in which the effect of the first wave of the COVID-19 pandemic on stress, depression and anxiety in patients diagnosed with different pathologies (panic disorder, social phobia and specific phobia) is analyzed. The results show a positive association between stressors and symptom load.

The introduction is correct; please add the following reference (page 3 line 33) which discusses the change in emotions that occurred in the first wave comparing two groups and points out the increase in depressive states due to the pandemic: Meléndez, J. C., Satorres, E., Reyes-Olmedo, M., Delhom, I., Real, E., & Lora, Y. (2020). Emotion recognition changes in a confinement situation due to COVID-19. Journal of Environmental Psychology, 72, 101518. https://doi.org/10.1016/j.jenvp.2020.101518

Measurements. Line 127. the authors note that: “For the latter, we changed the instructions of well-established psychometric tests for anxiety symptoms in a way that we ask for changes instead of intensity of symptoms.” It might be interesting for the authors to include an example of the modification.

Please provide more information on the Beck Depressive Inventory (Similar to Hamilton: internal reliability, scoring scale, cut-off points...). Please also provide information on how the Hamilton is scored: for example, "Each item is scored on a scale of 0 (not present) to 4 (severe)". Please add these characteristics to all scales used in the manuscript.

Although it is not necessary to include it in the manuscript, a question that arises in my mind is Why you did not use the same scale to assess depression during pre-pandemic and during pandemic? (BDI-II vs PHQ-9)

Regarding the modified questionnaires, I understand that what you have done is clinically correct, it is a good idea. However, it would be interesting if you could provide, if possible, information on the reliability of the modified questionnaires, this data would ensure that the modification would have been statistically correct.

In data analysis (page 9 line 166) you report: first hypothesis ..., second hypothesis. Please, when you state your research objectives at the end of the introduction, could you please match this numbering of hypotheses.

The results are correct. The explanatory figures provide the information in a satisfactory way and the regression analyses include all the necessary information.

The conclusions are correct and in accordance with the results.

6. PLOS authors have the option to publish the peer review history of their article (what does this mean?). If published, this will include your full peer review and any attached files.

Reviewer #1: No

Reviewer #2: No

Reviewer #3: No

---

## [Decision Letter · Decision Letter 1]

6 Jul 2022

PONE-D-21-37020R1Impact of COVID-19 pandemic related stressors on patients with anxiety disorders: A cross-sectional studyPLOS ONE

Dear Dr. Langhammer,

Thank you for submitting your manuscript to PLOS ONE. After careful consideration, we feel that it has merit but does not fully meet PLOS ONE’s publication criteria as it currently stands. Therefore, we invite you to submit a revised version of the manuscript that addresses the points raised during the review process.

Before acceptance, the manuscript needs some additional revisions, as suggested by reviewer 2.

We look forward to receiving your revised manuscript.

Kind regards,

Serena Scarpelli

Academic Editor

PLOS ONE

Journal Requirements:

Reviewers' comments:

Reviewer's Responses to Questions

**Comments to the Author**

1. If the authors have adequately addressed your comments raised in a previous round of review and you feel that this manuscript is now acceptable for publication, you may indicate that here to bypass the “Comments to the Author” section, enter your conflict of interest statement in the “Confidential to Editor” section, and submit your "Accept" recommendation.

Reviewer #2: (No Response)

Reviewer #3: All comments have been addressed

2. Is the manuscript technically sound, and do the data support the conclusions?

Reviewer #2: Yes

Reviewer #3: Yes

3. Has the statistical analysis been performed appropriately and rigorously? 

Reviewer #2: N/A

Reviewer #3: Yes

4. Have the authors made all data underlying the findings in their manuscript fully available?

Reviewer #2: No

Reviewer #3: No

5. Is the manuscript presented in an intelligible fashion and written in standard English?

Reviewer #2: Yes

Reviewer #3: Yes

6. Review Comments to the Author

Reviewer #2: Thank you for reviewing the comments. This has greatly helped to make the manuscript more understandable and of higher quality.

In the light of the revision, two important aspects arise that I would require to be answered.

1) The authors have stated that they have calculated a regression. My question at this point is: you included about 30 variables in the regression with a sample size of N = 47. This seems to me to be a very exploratory approach and my question is whether this number of variables is statistically justified?

Additionally, it would be great if in the methods section under statistical analyses it is still described that only significant variables are presented and all others are omitted here. It would also be interesting to know in which step which variable was included.

2) You wrote in your hypotheses/aims at the beginning (Introduction section) that you want to include health concerns, financial worries and pre-pandemic symptom severity as well as some socio-demographic variables to answer your question (regression analyses). But in the regression you find mainly variables of COVID-related stressors (number 12, 13, 5 and 16), which are not named in the hypothesis. Here I would like it to be clearer why the authors included these stressors in the analyses but did not mention them as variables to reach the aim at the end of the introduction.

Thank you so much for your effort and your great work!

Reviewer #3: The authors have correctly included the reviewers' suggestions and the manuscript has been improved.

7. PLOS authors have the option to publish the peer review history of their article (what does this mean?). If published, this will include your full peer review and any attached files.

Reviewer #2: No

Reviewer #3: No

---

## [Decision Letter · Decision Letter 2]

15 Jul 2022

Impact of COVID-19 pandemic related stressors on patients with anxiety disorders: A cross-sectional study

PONE-D-21-37020R2

Dear Dr. Langhammer,

We’re pleased to inform you that your manuscript has been judged scientifically suitable for publication and will be formally accepted for publication once it meets all outstanding technical requirements.

Kind regards,

Serena Scarpelli

Academic Editor

PLOS ONE

Additional Editor Comments (optional):

Reviewers' comments:

Reviewer's Responses to Questions

**Comments to the Author**

1. If the authors have adequately addressed your comments raised in a previous round of review and you feel that this manuscript is now acceptable for publication, you may indicate that here to bypass the “Comments to the Author” section, enter your conflict of interest statement in the “Confidential to Editor” section, and submit your "Accept" recommendation.

Reviewer #2: All comments have been addressed

2. Is the manuscript technically sound, and do the data support the conclusions?

Reviewer #2: Yes

3. Has the statistical analysis been performed appropriately and rigorously? 

Reviewer #2: Yes

4. Have the authors made all data underlying the findings in their manuscript fully available?

Reviewer #2: No

5. Is the manuscript presented in an intelligible fashion and written in standard English?

Reviewer #2: Yes

6. Review Comments to the Author

Reviewer #2: (No Response)

7. PLOS authors have the option to publish the peer review history of their article (what does this mean?). If published, this will include your full peer review and any attached files.

Reviewer #2: No

---

## [Editor Report · Acceptance letter]

9 Aug 2022

PONE-D-21-37020R2 

Impact of COVID-19 pandemic related stressors on patients with anxiety disorders: A cross-sectional study 

Dear Dr. Langhammer:

I'm pleased to inform you that your manuscript has been deemed suitable for publication in PLOS ONE. Congratulations! Your manuscript is now with our production department. 

Kind regards, 

on behalf of

Dr. Serena Scarpelli 

Academic Editor

PLOS ONE